# Influence of Short-Term Hyperenergetic, High-Fat Feeding on Appetite, Appetite-Related Hormones, and Food Reward in Healthy Men

**DOI:** 10.3390/nu12092635

**Published:** 2020-08-29

**Authors:** Alice E. Thackray, Scott A. Willis, David J. Clayton, David R. Broom, Graham Finlayson, Fernanda R. Goltz, Jack A. Sargeant, Rachel M. Woods, David J. Stensel, James A. King

**Affiliations:** 1National Centre for Sport and Exercise Medicine, School of Sport, Exercise and Health Sciences, Loughborough University, Loughborough LE1 3TU, UK; A.E.Thackray@lboro.ac.uk (A.E.T.); S.Willis2@lboro.ac.uk (S.A.W.); fernanda.reistenbachgoltz@rdsi.nestle.com (F.R.G.); D.J.Stensel@lboro.ac.uk (D.J.S.); 2National Institute for Health Research (NIHR) Leicester Biomedical Research Centre, University Hospitals of Leicester NHS Trust and University of Leicester, Leicester LE5 4PW, UK; 3School of Science and Technology, Nottingham Trent University, Nottingham NG1 8NS, UK; david.clayton@ntu.ac.uk; 4Centre for Sport, Exercise and Life Sciences, Coventry University, Coventry CV1 2DS, UK; ad5173@coventry.ac.uk; 5School of Psychology, Faculty of Medicine and Health, University of Leeds, Leeds LS2 9JT, UK; G.S.Finlayson@leeds.ac.uk; 6Diabetes Research Centre, University of Leicester, Leicester LE5 4PW, UK; jack.sargeant@leicester.ac.uk; 7Medical School, University of Lincoln, Lincoln LN6 7TS, UK; rawoods@lincoln.ac.uk

**Keywords:** appetite, food-reward, overfeeding, high-fat diet, energy balance, compensation

## Abstract

Short-term overfeeding may provoke compensatory appetite responses to correct the energy surplus. However, the initial time-course of appetite, appetite-related hormone, and reward-related responses to hyperenergetic, high-fat diets (HE-HFD) are poorly characterised. Twelve young healthy men consumed a HE-HFD (+50% energy, 65% fat) or control diet (36% fat) for seven days in a randomised crossover design. Mean appetite perceptions were determined during an oral glucose tolerance test (OGTT) before and after each diet. Fasted appetite perceptions, appetite-related hormones, and reward parameters were measured pre-diet and after 1-, 3- and 7-days of each diet. The HE-HFD induced a pre-to-post diet suppression in mean appetite during the OGTT (all ratings *p* ≤ 0.058, effect size (*d*) ≥ 0.31), and reduced the preference for high-fat vs. low-fat foods (main effect diet *p* = 0.036, *d* = 0.32). Fasted leptin was higher in the HE-HFD than control diet (main effect diet *p* < 0.001, *d* = 0.30), whilst a diet-by-time interaction (*p* = 0.036) revealed fasted acylated ghrelin was reduced after 1-, 3- and 7-days of the HE-HFD (all *p* ≤ 0.040, *d* ≥ 0.50 vs. pre-diet). Appetite perceptions and total peptide YY in the fasted state exhibited similar temporal patterns between the diets (diet-by-time interaction *p* ≥ 0.077). Seven days of high-fat overfeeding provokes modest compensatory changes in subjective, hormonal, and reward-related appetite parameters.

## 1. Introduction

Energy balance is determined by the relationship between energy intake and energy expenditure, and this topic attracts widespread scientific interest given its centrality in weight management. In set-point models of energy balance, homeostasis is a key concept, with adaptive responses to energy imbalance serving to attenuate deviations [1]. Importantly, owing to humans’ ‘survival physiology’, it is thought that adaptive responses to negative energy balance are stronger than those elicited by positive energy balance [2,3].

With relevance to eating behaviour, ‘appetite control’ and the notion of whether appetite is subject to regulation, has become a topic of increasing interest within energy balance research. Notably, extensive research has explored the impact of short- and long-term negative energy balance on appetite, appetite-related hormones (e.g., leptin, ghrelin, peptide YY (PYY)) and hedonic eating influences [4]. This work has shown that diet-induced energy deficits produce rapid compensatory appetite responses [5,6] that endure whilst energy stores remain depleted [7]. Such responses may explain why individuals find it difficult to achieve and maintain weight loss [8,9].

In contrast, whilst the biology of overfeeding is well characterised [10], the impact of positive energy balance on appetite control is not fully understood. For appetite perceptions, a few studies have examined the impact of sustained overfeeding (+30–50% energy) across 1–3 days [11,12,13,14]. The consensus from this work suggests that a short-term energy surplus is readily detected and elicits a compensatory reduction in perceived appetite. However, these studies employed balanced Western style diets, whereas overconsumption often occurs in response to palatable diets high in fat [15]. It remains unknown how appetite responds when a positive energy balance is provoked by high-fat diets.

When considering the mechanisms underpinning appetite responses to dietary interventions, an important distinction is made between ‘metabolic drivers’ and ‘hedonic’ influences [16]. Within homeostatic models, appetite-related hormones are recognised as key metabolic signals influencing appetite and eating behaviour [17]. These signals interact with cognitive and emotional factors that are conceptualised as reward-related (non-homeostatic) mediators of appetite and eating [18]. Existing evidence demonstrates that leptin (a chronic anorexigenic signal) increases in response to short-term periods of energy excess (1–5 days, ≥+25% energy) [3,19,20,21,22,23]. In contrast, the short-acting appetite-related signals (e.g., ghrelin and PYY) appear less responsive to short-term overfeeding [11,14,19,21,24]. Fewer studies have imposed hyperenergetic diets of high fat content (≥50%), and findings for appetite-related hormones within these are mixed [20,23,24,25]. Further work is required to clarify the responsiveness of appetite-related hormones to hyperenergetic, high-fat diets and to better characterise the time course of responses.

With the use of functional magnetic resonance imaging (fMRI), two studies have concluded that the neural responses to visual food cues are attenuated after 48 h of overfeeding (+30% energy) [13,26]. More recently, the Leeds Food Preference Questionnaire (LFPQ) has been employed as a more accessible technique to assess food preference and reward in laboratory settings [27]. To date, this technique has been used to characterise hedonic responses to negative energy balance [28,29], but no data are available regarding episodes of positive energy balance.

To extend the evidence base, this study sought to characterise the effect of short-term, high-fat overfeeding on appetite, appetite-related hormones, and hedonic influences of eating behaviour. Uniquely, we investigated the time-course of responses over seven days. Despite the imposition of extreme positive energy balance, we hypothesized that changes in subjective appetite perceptions would be subtle, whereas more robust responses would be seen for appetite-related hormones and reward-related eating parameters.

## 2. Materials and Methods

### 2.1. Ethical Approval and Participant Recruitment

This manuscript presents secondary endpoints from data collected in a study investigating hepatokine and metabolic responses to overfeeding; a detailed description of the study design and results of the primary study outcomes have been published [30]. After receiving institutional ethical approval, 12 healthy males were recruited into the study after providing written informed consent. Participants were healthy, did not smoke, were habitually active and reported being weight stable. The characteristics of the participants were as follows: mean (SD) age 24 (4) years; body mass 76.8 (4.5) kg; body mass index (BMI) 24.1 (1.5) kg/m^2^; body fat percentage 13.3 (3.0)%. The study was registered as a clinical trial (NCT03369145) before data collection began and was approved by Loughborough University research ethics committee (R17-P144).

### 2.2. Participant Screening

Participants were screened to determine eligibility for the study. During a two-hour laboratory visit, a medical history was taken, and assessments were made of the participants’ BMI, fasted blood glucose (finger-prick) and resting arterial blood pressure. Acceptability of food items to be provided during the study was checked and participants were familiarised with the procedures involved within the experimental trials. At the end of the visit, participants were provided with a hip-worn accelerometer (ActiGraph GT3X, ActiGraph Corp, USA) to assess habitual physical activity over the subsequent week. A three-day weighed food record (two weekdays and one weekend day) was also completed during this time to estimate baseline habitual energy and macronutrient intake.

### 2.3. Study Design and Procedures

This study used a randomised-counterbalanced crossover design, where each participant completed two, seven-day dietary interventions separated by a three-week washout period: (1) hyperenergetic, high-fat diet (HE-HFD); and (2) control diet (Figure 1).

Within each dietary intervention, participants attended four laboratory visits. These occurred at baseline (pre-diet) and after one, three and seven days. Participants attended each visit after an overnight fast (≥10 h) and having abstained from caffeine, alcohol, and exercise for 24 h. During each visit, a fasted venous blood sample was obtained, and fasted appetite ratings were assessed using 100 mm visual analogue scales (hunger, fullness, desire to eat, prospective food consumption). Participants also completed the LFPQ to assess hedonic factors known to influence eating behaviour. At the baseline (pre-diet) and seven-day visits, participants additionally underwent a 2 h oral glucose tolerance test (OGTT). During this test, appetite ratings were assessed at 0 h (fasted), 1 h and 2 h after the consumption of 82.5 g of dextrose monohydrate (MyProtein, Cheshire, UK) and the mean response across 2 h was calculated for each rating (note: insulin and glucose data have been published previously [30]). Within each dietary intervention, participants were instructed not to alter their habitual physical activity levels and compliance with this request was confirmed via accelerometry (data published previously [30]).

### 2.4. Dietary Interventions

Within the HE-HFD intervention, participants consumed 150% of their estimated energy requirements. To determine the required dietary energy, participants’ daily energy needs were calculated using established equations for resting metabolic rate (resting energy expenditure = 10 × weight (kg) + 6.25 × height (cm) − 5 × age (years) + 5) [31]. This value was multiplied by a physical activity correction factor that represents ‘moderate’ habitual activity levels (1.7) and uplifted by 10% to adjust for the thermic effect feeding. The HE-HFD consisted of approximately 65% fat, 21% carbohydrate and 14% protein. Of the total energy content, 32% was derived from saturated fatty acids (SFA), whilst 26% and 8% were derived from mono- (MUFA) and poly-unsaturated fatty acids (PUFA), respectively. Within the HE-HFD, all foods and energy-containing drinks were prepared by the research team. Participants consumed all foods provided to them and no additional energy-containing food or drinks. Dietary compliance was verbally clarified at study visits.

Within the control diet, participants consumed their habitual diet for seven days. To facilitate compliance, a three-day weighed food record (two weekdays and one weekend day) was completed which was subsequently cross-checked against each participant’s baseline diet record.

### 2.5. Leeds Food Preference Questionnaire

At baseline (pre-diet), one, three and seven days, participants completed the LFPQ, which is a validated laptop-based procedure that measures food preference and reward [32]. The LFPQ provides measures of wanting and liking for an array of food images that vary in fat content and taste. The conduct and analysis of this questionnaire have been described previously [33]. In the present study, bias scores for fat appeal and sweet appeal were obtained by subtracting the low-fat scores from the high-fat scores and the savory scores from the sweet scores, respectively.

### 2.6. Biochemical Analyses

Blood samples were collected into ice-cooled potassium EDTA monovettes (Sarstedt, Leicester, UK) and were spun immediately in a refrigerated centrifuge (Heraeus Labofuge 400R, Thermo Fisher Scientific, Waltham, MA, USA) at 4 °C for 10 min (2383× *g*). Plasma was then removed and aliquoted for storage at −80 °C. To prevent protein degradation by cysteine-proteases, samples (monovettes) for acylated ghrelin were pre-treated with a 50 µL solution comprising 0.1 M phosphate buffered saline, *P*-hydroxymeruribenzoic acid and 10 M sodium hydroxide. After centrifugation, plasma was acidified with 1 M hydrochloric acid and re-centrifuged for five min. Commercially available enzyme-linked immunosorbent assays (ELISAs) were used to measure plasma concentrations of leptin (DLP00, R & D Systems, Oxford, UK; CV_intra_ 6.8%), acylated ghrelin (97751 SCETI, Tokyo, Japan; CV_intra_ 4.3%) and total PYY (EZHPPPT66K, Millipore, Watford, UK; CV_intra_ 2.4%). All samples from each participant were analysed on the same assay plate to reduce variation.

### 2.7. Statistical Analyses

The data reported in this manuscript are secondary outcomes from a previous trial [30], and consequently, the sample size was not informed by a formal power calculation.

Data were analysed using the software package IBM SPSS Statistics for Windows version 25.0 (IBM Corporation, New York, NY, USA). The model residuals for all outcome variables were explored using histograms. Appetite perceptions and food preference and reward scores were normally distributed and presented as mean (SD). Concentrations of leptin, acylated ghrelin and total PYY demonstrated a positively skewed distributed and were natural log transformed. The model residuals were re-checked for these outcomes to confirm parity to a Gaussian distribution before analysis and the variables are presented descriptively as geometric mean (95% confidence interval (95% CI)).

Between-diet differences in pre-diet study outcomes were examined using linear mixed models with diet (control diet and HE-HFD) as the sole fixed factor. Differences in fasted appetite perceptions, appetite-related hormones and food preference and reward scores between the two diets across the seven-day intervention were examined using linear mixed models with two fixed effects (diet: control diet and HE-HFD; and time: pre-diet, 1 day, 3 day and 7 day). Linear mixed models including diet (control diet and HE-HFD) and time (pre-diet and 7 day) as fixed effects were used to examine between-diet differences in mean appetite perceptions during the OGTT at the pre-diet and seven day timepoints.

Absolute standardised effect sizes (Cohen’s *d*) were calculated to supplement important findings, and thresholds of 0.2, 0.5, and 0.8 describe small, moderate, and large effects, respectively. For normally distributed variables, pairwise comparisons are based on mean differences and the respective 95% CI of the mean absolute difference. For log transformed variables, pairwise comparisons are based on the ratio of geometric means and 95% CI for the ratio of geometric means. Interpretation of the data is based on the 95% CI and effect sizes rather than more conventional dichotomous hypothesis testing [34].

## 3. Results

### 3.1. Ratings of Perceived Appetite

The 95% CI for the mean difference in pre-diet fasted ratings of perceived appetite between the control diet and the HE-HFD overlapped zero (all *p* ≥ 0.428) (Table 1). All standardised effect sizes were trivial (*d* ≤ 0.15) except for the small-to-moderate effect size for pre-diet fasted fullness (*d* = 0.31) (Table 1). Fasted ratings of perceived hunger, fullness, desire to eat and prospective food consumption (PFC) were also similar between the two diets over time (all main effect diet *p* ≥ 0.183; main effect time *p* ≥ 0.085; diet-by-time interaction *p* ≥ 0.373) (Figure 2).

There were no main effects of diet (all *p* ≥ 0.187) or time (all *p* ≥ 0.353) for the mean ratings of perceived appetite during the OGTT, but diet-by-time interactions were identified for hunger (*p* = 0.030), fullness (*p* = 0.004), desire to eat (*p* = 0.022) and PFC (*p* = 0.053) (Figure 3). Specifically, the HE-HFD elicited a pre-to-post diet reduction in mean ratings of hunger, desire to eat and PFC, and an increase in mean ratings of fullness during the OGTT (all *p* ≤ 0.058, *d* ≥ 0.31) (Figure 3). In the control diet, the 95% CI for the pre-to-post diet change in mean appetite ratings during the OGTT all overlapped zero, but the magnitude of reduction in mean fullness was moderate-to-large (*p* = 0.077, *d* = 0.67) (Figure 3B).

### 3.2. Appetite-Related Hormones

All 95% CIs for the mean difference in pre-diet fasted concentrations of leptin, acylated ghrelin and total PYY between the control diet and the HE-HFD overlapped zero (all *p* ≥ 0.165) (Table 1, Figure 4). Standardised effect sizes were trivial for fasted leptin (*d* = 0.09) and acylated ghrelin (*d* = 0.07) and small-to-moderate for fasted total PYY (*d* = 0.38) (Table 1). The removal of an outlier for leptin, which was >3.5 residual SDs from the mean predicted value [35], did not affect the interpretation of the main effect of diet (control diet: 3051 (1922, 4845) pg/mL, HE-HFD: 3441 (2167, 5464) pg/mL; *p* = 0.353, *d* = 0.17).

Fasted plasma concentrations of leptin, acylated ghrelin and total PYY during the seven-day control diet and HE-HFD are presented in Figure 4. A main effect of diet (*p* < 0.001), time (*p* = 0.034), but not a diet-by-time interaction (*p* = 0.205), was identified for leptin, with higher fasted plasma leptin concentrations in the HE-HFD compared with the control diet (ratio difference (95% CI): 28 (17, 41)%; *d* = 0.30). Exclusion of the outlier highlighted previously for leptin successfully improved the non-normal distribution of the model residuals and a sensitivity analysis excluding the outlier yielded comparable findings to the model including all participants (main effect diet *p* < 0.001; main effect time *p* = 0.050; diet-by-time interaction *p* = 0.192).

A main effect of diet (*p* < 0.001), diet-by-time interaction (*p* = 0.036), but not a main effect of time (*p* = 0.187), was identified for acylated ghrelin. Post hoc analysis of the diet-by-time interaction revealed higher fasted acylated ghrelin concentrations in the control diet at one day compared with three day (ratio difference (95% CI): 11 (−0.4, 23)%; *p* = 0.060, *d* = 0.46), whereas concentrations in the HE-HFD were lower at one, three and seven days compared with pre-diet (ratio difference (95% CI): one day vs. pre-diet −12 (−21, −3)%, *p* = 0.015, *d* = 0.60; three day vs. pre-diet −14 (−22, −4)%, *p* = 0.007, *d* = 0.67; seven day vs. pre-diet −10 (−19, −1)%, *p* = 0.040, *d* = 0.50).

Fasted PYY concentrations changed over time (*p* = 0.029) but the magnitude and the temporal pattern were similar between the control diet and the HE-HFD (main effect diet *p* = 0.866; diet-by-time interaction *p* = 0.077).

### 3.3. Food Preference and Reward

The 95% CI for the mean difference in pre-diet food preference and reward scores between the control diet and the HE-HFD overlapped zero (all *p* ≥ 0.333) (Table 2). All standardised effect sizes were trivial (*d* ≤ 0.17) apart from the small-to-moderate effect size for pre-diet explicit liking taste bias (*d* = 0.26). A main effect of diet for relative preference fat appeal bias revealed lower scores in the HE-HFD than the control diet (mean difference (95% CI): −2.1 (−4.0, −0.1) AU; *p* = 0.036, *d* = 0.32) (Table 2). The main effect of diet for explicit liking fat appeal bias was not statistically significant (*p* = 0.055), but values were lower in the HE-HFD than the control diet (mean difference (95% CI) −3.1 (−6.3, 0.1) mm; *d* = 0.31) (Table 2). No other main effects of diet (all *p* ≥ 0.086), time (all *p* ≥ 0.141) or diet-by-time interactions (all *p* ≥ 0.148) were identified for any of the food preference and reward scores (Table 2).

## 4. Discussion

The primary findings from this study were that circulating fasted leptin and acylated ghrelin concentrations responded quickly (within 24 h) to the HE-HFD in directions expected to help correct the energy imbalance. Appetite perceptions were not altered in the fasted state but exhibited similar compensatory responses during the OGTT on day seven. Finally, within 24 h, the HE-HFD reduced the participants’ desire and motivation to consume high-fat foods. These alterations in subjective, hormonal, and hedonic indicators of appetite were relatively modest and of small-to-moderate magnitude, despite the considerable energy surplus delivered in our overfeeding model.

In this study, the HE-HFD did not alter any fasted subjective appetite ratings during the seven-day diet. Conversely, when mean responses were calculated across the 2 h OGTT, a small-to-moderate decrease in hunger, desire to eat and PFC, and a moderate-to-large increase in fullness was detected. These results suggest that the HE-HFD elicited compensatory appetite responses that emerged in response to glucose intake after several days of overconsumption. A few previous studies have investigated subjective appetite responses to one [11,22], two [26] and three [12,14] days of energy excess. With one exception [11], short-term overfeeding (+30–50% energy) led to compensatory appetite changes that would be expected to help correct the energy balance perturbation. In the exceptional case, where one day of overfeeding (+50% energy) had no impact on fasted or meal-related appetite, it is relevant that energy intake was covertly manipulated [11]. This may highlight the importance of psychological factors in the early appetite responses to overfeeding.

The assessment of appetite using fasted ratings across the course of this study is a potential limitation as single assessments of appetite are less sensitive than average measurements taken over an extended period [36]. Instead, the characterisation of serial responses to a standardised meal challenge is regarded as a more sensitive and ecologically valid alternative. However, it has been reported that one day of overfeeding (+50% energy) a balanced macronutrient diet caused a marked reduction in fasted hunger (24%) and increase in post-meal satiety (74%) the next morning [37]. Similarly, overfeeding (+40–50% energy) a balanced macronutrient diet over three days was shown to reduce hunger and increase satiety, when calculated from pre-meal measurements averaged over the study duration [12,14]. These data suggest that if a robust response is genuinely present, fasted appetite assessments can detect subjective responses to dietary manipulation; however, these results contrast those documented in the fasted state at one and three days in this study.

The contrasting appetite (fasting) findings between this study and previous investigations may be linked to the composition of the experimental diets. Specifically, balanced Western style diets (~50% carbohydrate, ~30% fat and ~20% protein) were implemented in the aforementioned studies, whereas we employed a diet especially high in fat (~65%) and low in carbohydrate (~21%). This difference in dietary protocol suggests that macronutrient content may be important in determining any compensatory appetite responses to overfeeding. This notion is consistent with the concepts of ‘passive overconsumption’ [15] and ‘high-fat hyperphagia’ [38]. Specifically, the high energy density and palatability of high-fat diets, combined with their weak influence on satiety, may explain why our HE-HFD did not affect appetite until day seven. However, given perceived appetite was altered in response to the OGTT (but not in the fasted state) on day seven of the HE-HFD, it is possible that earlier detection of appetite changes to overfeeding may have been precluded without an OGTT at one and three days.

Within this study, fasted blood samples were taken after one, three and seven days to characterise appetite-related hormone responses to the HE-HFD. Our data show that circulating leptin concentrations increased rapidly (within 24 h) in response to high-fat overfeeding and remained elevated, at approximately the same level, whilst the diet persisted. This short-term response is consistent with previous overfeeding studies that described leptin responses during 1–5 days of overfeeding (≥+25% energy) balanced macronutrient [3,19,21,22] or high-fat [20,23] diets. In this study, it is notable that circulating leptin concentrations did not increase further at three or seven days despite the continuation of the HE-HFD. This may suggest that a ceiling for enhanced leptin production was reached in response to short-term overfeeding.

A limitation of this study is that we cannot provide mechanistic insight; however, it is possible that augmented circulating insulin was responsible for stimulating leptin production from adipocytes [39]. We have reported previously that fasted circulating insulin, and insulin responses to the OGTT, were elevated in response to the HE-HFD [30]. The physiological relevance of the leptin response to overfeeding is not clear. Given leptin has appetite suppressive effects, the increase with overfeeding may represent a compensatory response seeking to maintain energy homeostasis. However, others have questioned this assumption [40], and argue that the satiety-promoting effects of exogenous leptin may only be apparent in conditions of hypoleptinemia [41]. Further mechanistic studies are needed to understand the physiological relevance of changes in circulating leptin in response to overfeeding.

In addition to leptin, we also measured acylated ghrelin and total PYY responses to the HE-HFD. Although distinct roles in the long-term regulation of energy balance have been described [42,43], these peptides are foremost thought of as short-term (meal-to-meal) modulators of appetite and eating [17]. Specifically, acylated ghrelin is an orexigenic peptide harboring various roles in the cephalic phase of digestion [44,45]. Conversely, PYY promotes satiety and efficient nutrient absorption via the ileal break mechanism [46,47]. In this study, fasted acylated ghrelin was suppressed rapidly after one day of high-fat overfeeding and remained lower throughout the HE-HFD, whereas fasted PYY was unresponsive to the HE-HFD.

The acylated ghrelin findings in this study contrast the balance of previous investigations demonstrating that short-term (1–7 days) overfeeding (+25–60% energy) of balanced and high-fat diets do not affect total [14,19,20] or acylated [11,21,24] ghrelin concentrations. The higher energetic and fat excess imposed by our diet over seven days is the most likely explanation for this discrepant outcome. However, it is notable that fasted acylated ghrelin concentrations remained unchanged after seven days of a similar HE-HFD [24]. This disparity likely reflects differences in the ELISA utilized and/or increased statistical power of a within-measures crossover design in the current study. Nevertheless, the suppression in fasted acylated ghrelin did not occur with congruent changes in fasted appetite perceptions, and the overall reduction in concentrations was modest despite the extreme overfeeding challenge. This may question the physiological importance of our acylated ghrelin finding.

Previous studies have shown that fasted and postprandial total PYY are unresponsive to 1–3 days of balanced macronutrient energy excess (+40–50% energy) [11,14,22], whereas more prolonged overfeeding lasting 5–7 days elicited modest increases in fasted total PYY (+50–70% energy) [20,48]. Given the similarity between study protocols, it is not clear why circulating PYY was not elevated in this study after seven days. Nonetheless, as PYY is a satiety-related peptide, additional research is needed to determine whether short-term overfeeding modulates PYY responses to nutrient challenges.

In this study, the HE-HFD modulated LFPQ parameters associated with the hedonic control of eating. Specifically, the HE-HFD induced a reduction in the relative preference for high-fat vs. low-fat foods. Additionally, lower implicit wanting and explicit liking of high-fat vs. low-fat food items were identified (although these outcomes were not statistically significant). Each of these responses were visually apparent after one day of the HE-HFD and persisted throughout the seven-day diet. The overconsumption of high-fat foods therefore appears to quickly attenuate the expected pleasure of, and motivation to obtain, foods that are high in fat.

When seeking to situate these findings, it is important to note that although many studies have examined the effects of short-term energy deficits on food reward (food restriction and exercise) [6,29,49], the influence of positive energy balance (overfeeding) has received substantially less attention. Furthermore, reward-related outcomes have only been assessed using fMRI, and therefore direct comparison with our results is not possible. Nonetheless, data from two experiments have shown that two days of overfeeding (+30% energy) reduces the neural response to visual food cues in individuals who are lean [13,26]. In contrast, negative energy balance, when induced by extended overnight fasting (vs. the fed-state), increased the activation of reward-related brain regions (ventral striatum, amygdala, anterior insula, medial and lateral orbitofrontal cortex) when participants were presented with high- vs. low-calorie foods [6].

To date, no studies have assessed the impact of overfeeding on food reward using the LFPQ. Alternatively, two studies have employed the LFPQ to examine the acute effects of energy restriction on food reward. Thivel et al. [29] recently showed that one day of fasting increased the preference and unconscious motivation (implicit wanting) to consume high-fat vs. low-fat foods. Others reported that one day of fasting increased the explicit liking of high-fat savory and low-fat sweet foods; and the explicit wanting of foods in all LFPQ categories (high-fat savory, high-fat sweet, low-fat savory, low-fat sweet) [28]. Collectively, the emerging evidence suggests that negative and positive energy balance may have opposing effects on food reward that could facilitate energy homeostasis. It is important to note, however, that despite the excessive amount of energy and fat consumed in our HE-HFD, the resulting absolute changes in LFPQ parameters were subtle. Our results may therefore indicate that the magnitude of early hedonic responses to high-fat overfeeding is slight, and possibly less robust than responses to underfeeding/negative energy balance. Further head-to-head studies are needed to test this hypothesis, and to investigate how macronutrient content influences responses to overfeeding.

The robust standardisation of dietary and experimental procedures is a key strength of this study; however, it is recognised that the experimental design could have been strengthened by the provision of all foods to participants in both trial arms (not just in the HE-HFD arm). A notable limitation is that subjective appetite perceptions and appetite-related hormones were examined using single fasting assessments, which are less sensitive than averaged responses over the course of standardised meal challenges. Finally, a homogenous sample of young, healthy males were recruited which limits the generalisability of our data.

## 5. Conclusions

In conclusion, high-fat overfeeding (+50% energy, 65% fat) elicits early, albeit subtle, changes in metabolic (elevated leptin and reduced acylated ghrelin) and hedonic (reduced relative preference, explicit liking and implicit wanting for high-fat vs. low-fat foods) mediators of appetite within 24 h. These responses were shown to persist at three and seven days whilst the dietary challenge endured. Conversely, fasted appetite perceptions were unaltered with overfeeding but exhibited compensatory changes during the OGTT after seven days. The modest alterations observed, and the potential disconnect between perceived appetite and purported modulating factors, challenges the extent that subjective, physiological, and hedonic appetite signals can counteract short periods of extreme energy surplus.

## Figures and Tables

**Figure 1 nutrients-12-02635-f001:**
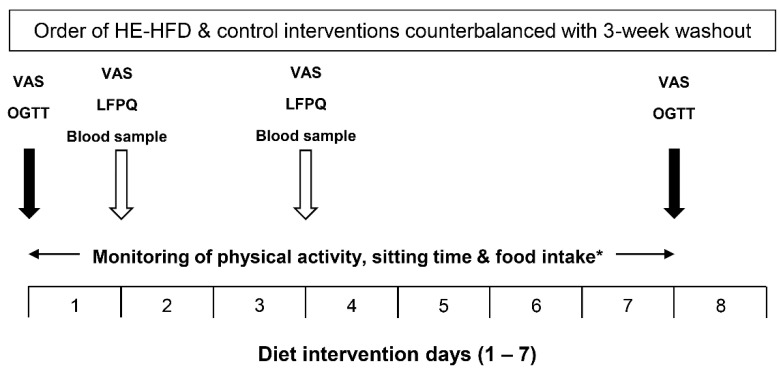
Schematic illustration of the main trial protocol. HE-HFD, hyperenergetic high-fat diet; VAS, visual analogue scale; OGTT, oral glucose tolerance test; LFPQ, Leeds Food Preference Questionnaire. * Physical activity, sitting time and food intake data collected during the control diet and the HE-HFD have been reported previously [30].

**Figure 2 nutrients-12-02635-f002:**
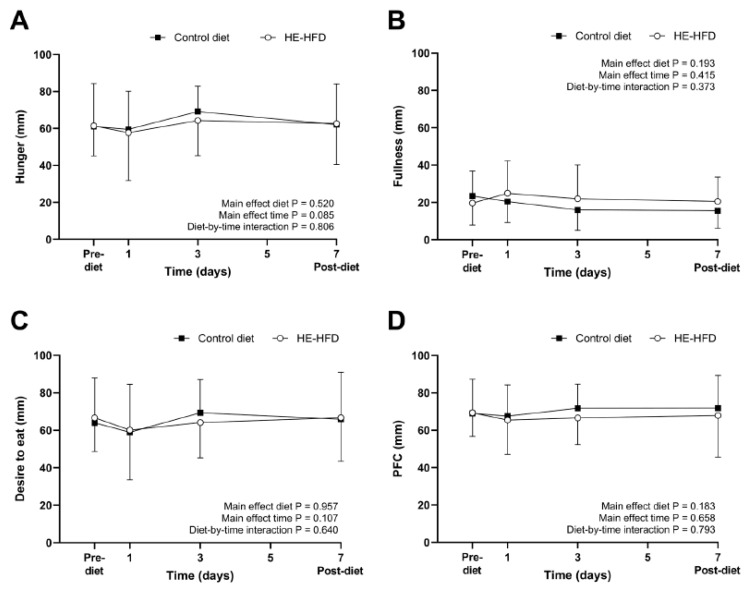
Fasted ratings of perceived (**A**) hunger, (**B**) fullness, (**C**) desire to eat, and (**D**) prospective food consumption (PFC) during the seven-day control diet and hyperenergetic high-fat diet (HE-HFD) in 12 healthy men. Values are mean (SD).

**Figure 3 nutrients-12-02635-f003:**
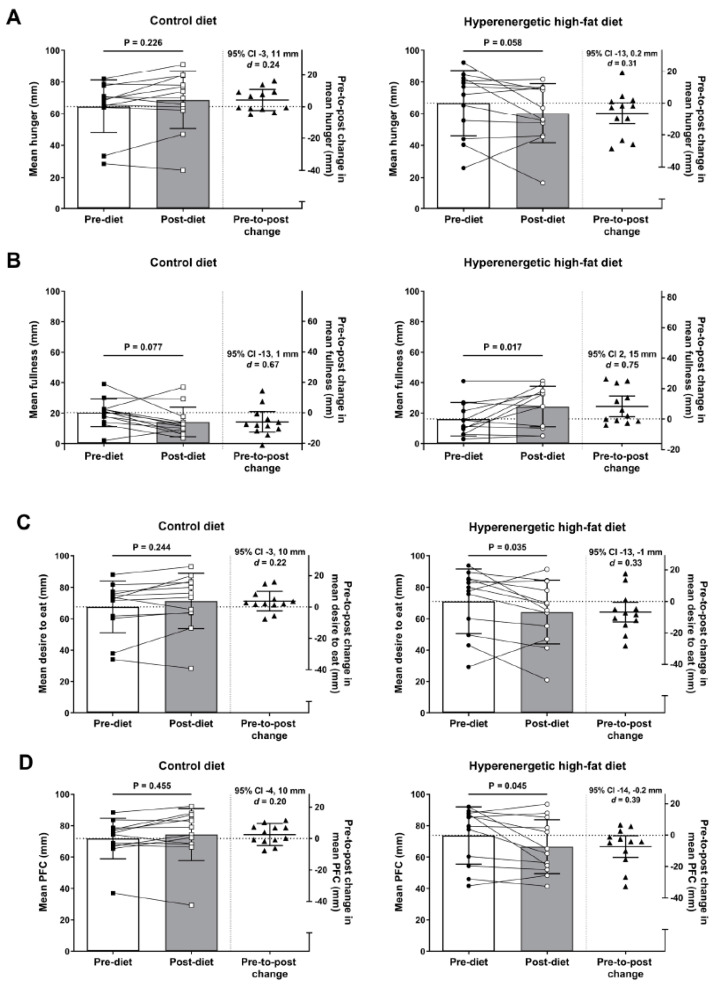
Mean ratings of perceived (**A**) hunger, (**B**) fullness, (**C**) desire to eat, and (**D**) prospective food consumption (PFC) during the oral glucose tolerance test before (pre-diet) and after (post-diet) the seven-day control diet (left panels) and hyperenergetic high-fat diet (HE-HFD) (right panels) in 12 healthy men. The left-hand y axis displays the pre-diet and post-diet appetite data separately: bars and error bars represent the mean (SD), and data points (squares ■, □ and circles ●, ○) and connecting lines represent the pre- and post-diet individual data values. The right-hand y axis displays the pre-to-post diet change in appetite: the horizontal lines represent the mean difference (95% confidence interval (CI) of the mean absolute difference), and the triangles (▲) indicate the pre-to-post change for each participant. Displayed *p* values, 95% CI (of the mean difference) and effect sizes (Cohen’s *d*) are from the post-hoc analysis of diet-by-time interactions for hunger *p* = 0.030, fullness *p* = 0.004, desire to eat *p* = 0.022, and PFC *p* = 0.053.

**Figure 4 nutrients-12-02635-f004:**
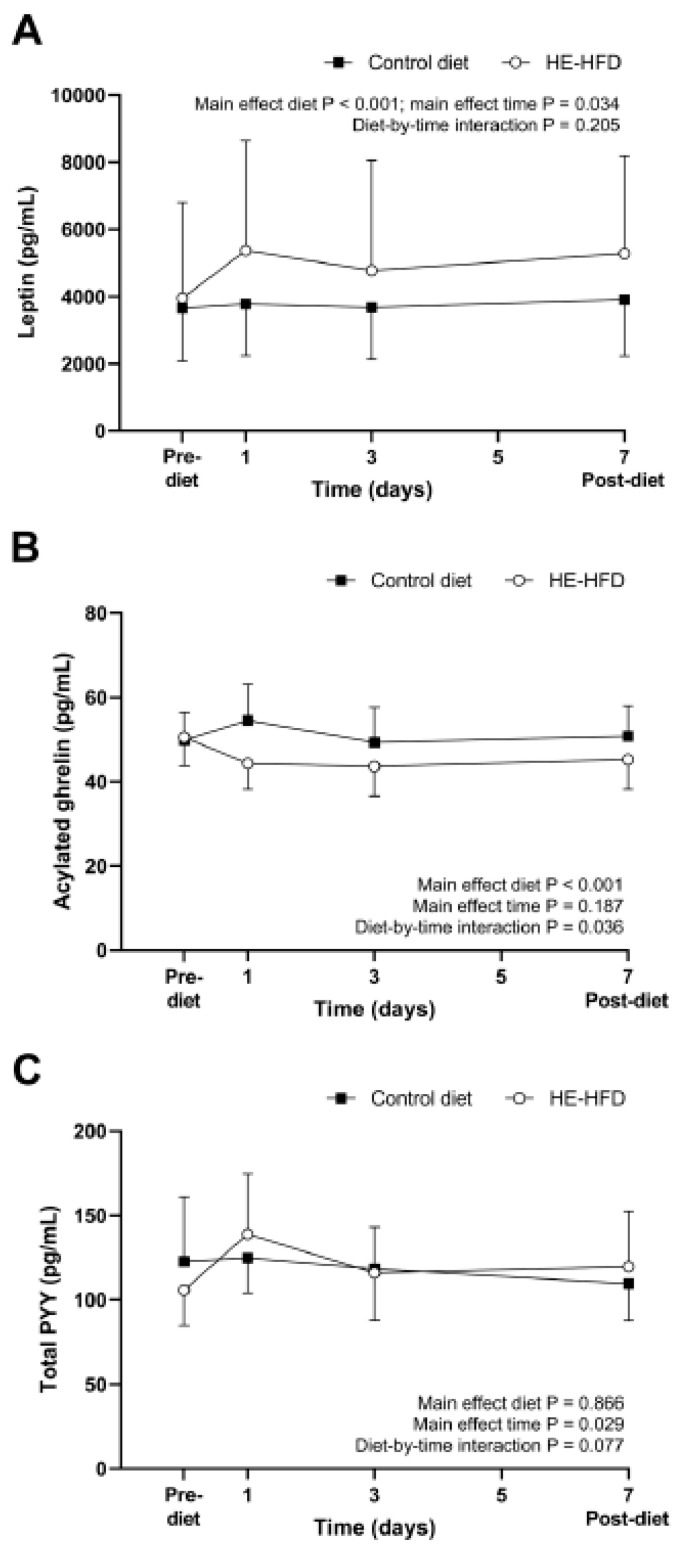
Fasted plasma concentrations of (**A**) leptin, (**B**) acylated ghrelin and (**C**) total peptide YY (PYY) during the seven-day control diet and hyperenergetic high-fat diet (HE-HFD) in 12 healthy men. Values are geometric mean (95% confidence interval).

**Table 1 nutrients-12-02635-t001:** Pre-diet fasted ratings of perceived appetite and appetite-related hormone concentrations during the control diet and hyperenergetic high-fat diet.

Outcome	Control Diet	HE-HFD	Main Effect Diet HE-HFD vs. Control Diet		
			Mean Difference (95% CI) ^1^	*p* Value	Effect Size (*d*)
**Ratings of Perceived Appetite**					
Hunger (mm)	61 (16)	61 (23)	0.3 (−9, 10)	0.939	0.02
Fullness (mm)	24 (13)	20 (12)	−4 (−14, 6)	0.428	0.31
Desire to eat (mm)	64 (15)	67 (21)	3 (−7, 13)	0.556	0.15
PFC (mm)	69 (12)	69 (18)	0.3 (−8, 9)	0.933	0.02
**Appetite-related hormones**					
Leptin (pg/mL)	3653 (2114, 6310)	3956 (2290, 6833)	8% (−17, 41%)	0.520	0.09
Acylated ghrelin (pg/mL)	49.8 (44.2, 56.2)	50.5 (44.8, 56.9)	1% (−4, 8%)	0.623	0.07
Total PYY (pg/mL)	122.8 (96.7, 155.8)	105.8 (83.4, 134.3)	−14% (−31, 7%)	0.165	0.38

Values for ratings of perceived appetite are mean (SD) for *n* = 12. Values for leptin, acylated ghrelin and total PYY are geometric mean (95% CI) for *n* = 12 and statistical analyses are based on log transformed data. Data were analysed using linear mixed models with diet (control diet or HE-HFD) included as a fixed factor. ^1^ For normally distributed data, values represent the mean absolute difference (95% CI of the mean absolute difference between the diets). For log transformed data, values represent the ratio of geometric means (95% CI for the ratio of geometric means between the diets). CI, confidence interval; HE-HFD, hyperenergetic high-fat diet; PFC, prospective food consumption; PYY, peptide YY.

**Table 2 nutrients-12-02635-t002:** Measures of relative preference, implicit wanting, explicit wanting and explicit liking during the seven-day control diet and hyperenergetic high-fat diet.

Outcome	Pre-Diet	1 Day	3 Day	7 Day	Main Effect Diet HE-HFD vs. Control Diet			Diet-by-Time Interaction *p* Value
					Mean Difference (95% CI) ^1^	*p* Value	Effect Size (*d*)	
**Relative Preference Fat Appeal Bias (AU)**								
Control diet	10.1 (7.7)	10.8 (5.7)	12.0 (6.1)	10.5 (6.7)	−2.1 (−4.0, −0.1)	0.036	0.32	0.470
HE-HFD	10.7 (8.3)	8.0 (8.6)	8.9 (7.7)	7.3 (7.6)				
**Relative Preference Sweet Appeal Bias (AU)**								
Control diet	12.8 (13.4)	10.3 (13.0)	12.1 (9.7)	7.9 (12.1)	−0.3 (−2.3, 1.7)	0.788	0.09	0.558
HE-HFD	10.7 (12.8)	9.4 (15.3)	9.2 (13.3)	9.6 (13.4)				
**Implicit Wanting Fat Appeal Bias (AU)**								
Control diet	25.5 (17.7)	26.8 (13.1)	28.3 (13.3)	23.3 (14.5)	−4.9 (−10.5, 0.7)	0.086	0.33	0.441
HE-HFD	27.6 (19.3)	16.5 (24.3)	21.4 (18.4)	19.0 (19.1)				
**Implicit Wanting Sweet Appeal Bias (AU)**								
Control diet	32.7 (40.8)	27.3 (37.0)	32.3 (22.7)	21.0 (30.9)	−0.5 (−6.4, 5.4)	0.863	0.07	0.228
HE-HFD	27.5 (32.4)	19.9 (39.8)	26.3 (35.2)	29.3 (42.1)				
**Explicit Wanting Fat Appeal Bias (mm)**								
Control diet	7.8 (10.7)	4.6 (8.1)	6.7 (9.5)	7.3 (9.5)	−1.9 (−4.8, 1.0)	0.197	0.19	0.148
HE-HFD	7.0 (6.0)	7.2 (9.0)	4.8 (12.0)	0.6 (10.7)				
**Explicit Wanting Sweet Appeal Bias (mm)**								
Control diet	14.6 (14.2)	14.8 (14.6)	13.7 (16.6)	11.0 (18.1)	−0.3 (−3.2, 2.5)	0.822	0.07	0.973
HE-HFD	13.6 (18.4)	13.8 (22.0)	11.4 (17.0)	11.2 (14.7)				
**Explicit Liking Fat Appeal Bias (mm)**								
Control diet	8.9 (12.3)	10.2 (8.1)	8.8 (10.3)	9.5 (8.8)	−3.1 (−6.3, 0.1)	0.055	0.31	0.842
HE-HFD	8.2 (7.2)	5.8 (9.6)	6.1 (12.2)	5.4 (10.8)				
**Explicit Liking Sweet Appeal Bias (mm)**								
Control diet	18.9 (21.0)	18.3 (19.0)	19.1 (20.1)	15.5 (22.3)	−3.5 (−8.3, 1.3)	0.150	0.21	0.934
HE-HFD	13.8 (17.5)	15.0 (20.0)	11.6 (15.4)	14.5 (16.6)				

Values are mean (SD) for *n* = 12. Data were analysed using linear mixed models with diet (control diet, HE-HFD) and time (pre-diet, 1 day, 3 day, 7 day) included as fixed factors. ^1^ Mean difference and 95% CI of the mean absolute difference between the control diet and the HE-HFD. CI, confidence interval; HE-HFD, hyperenergetic high-fat diet.

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
