# Peer review of "Influence of Short-Term Hyperenergetic, High-Fat Feeding on Appetite, Appetite-Related Hormones, and Food Reward in Healthy Men"

_nutrients, 2020, doi:10.3390/nu12092635_

Round 1

Reviewer 1 Report

  1. Why did you choose 7 days as the intervention?
  2. Why did you choose 3 weeks as the washout between interventions?
  3. Power calculation missing from 2.7 Statistical analyses. Please add how you came up with the need to have 12 men.
  4. Figure 3 very confusing. Is there any comparison between control diet and hyperenergetic diet or is it a straight presentation of pre and post for each diet?
  5. I think the authors are a bit enthusiastic about the results of the Leeds food Preference questionnaire presented in table 2. The conclusion is a little strong saying that the HE-HFD evoked a rapid reduction in the participants' desire and motivation to consume high-fat foods.

Author Response

Response to reviewers comments

We thank the reviewers for giving their time to critically appraise our paper. We hope that we have interpreted the comments accurately and that our responses are satisfactory.

Reviewer 1

Reviewer comment 1: Why did you choose 7 days as the intervention?

Authors’ response: As we highlighted in our methods (page 2, lines 82-84), these appetite-related data are secondary outcomes from a previous study which investigated the effects of HE-HFD on liver-related metabolism. The study duration was therefore fixed and selected based on the duration of overfeeding expected to influence the primary outcomes in the study. However, we made the decision to include appetite-related outcomes a priori (and included them in the clinical trials registration) because we saw the value in examining appetite responses to overfeeding across a 7-day time frame. Notably, within the available literature, there are previous studies investigating similar extreme overfeeding paradigms with very short durations of observation (1 to 3 days) and therefore a novel aspect of this study is the examination of time-course responses over a more prolonged period.

Reviewer comment 2: Why did you choose 3 weeks as the washout between interventions?

Author’s response: A washout period of 3 weeks between interventions is consistent with that utilised in other studies with similar dietary interventions (Lovejoy et al. (2002) https://doi.org/10.2337/diacare.25.8.1283; Lundsgaard et al. (2017) https://doi.org/10.1016/j.molmet.2016.11.001; Samms et al. (2017) https://doi.org/10.1210/jc.2017-01257). In our previous publication (Willis et al. (2020) https://doi.org/10.1093/jn/nxz333), reporting the primary outcomes from this dataset, we found no carryover effects from the HE-HFD intervention (into the control trial for the 50% of participants undertaking trials in this order).

We are also confident that this duration was sufficient to prevent carry over effects for our appetite-related outcomes reported in the present manuscript. Specifically, retrospective analysis of the six participants who completed the HE-HFD first revealed that the 95% CI for the mean difference in pre-diet values between the two interventions overlapped zero for all appetite outcomes (all P ≥ 0.118).

Reviewer comment 3: Power calculation missing from 2.7 Statistical analyses. Please add how you came up with the need to have 12 men.

Authors’ response: As the data presented in this manuscript were secondary outcomes from a previous trial, a formal power calculation was not performed. We have added a sentence to the statistical analyses section of our methods to make this clear (page 4, lines 157-158). This reads: ‘The data reported in this manuscript are secondary outcomes from a previous trial [30], and consequently, the sample size was not informed by a formal power calculation.’

Reviewer comment 4: Figure 3 very confusing. Is there any comparison between control diet and hyperenergetic diet or is it a straight presentation of pre and post for each diet?

Authors’ response: We choose to present the mean appetite perceptions from the OGTT in the format shown in Figure 3 to display the mean pre-diet data, mean post-diet data, and the mean pre-to-post diet change for the two interventions. The data for the two diets are displayed in separate panels for clarity and we were keen to display individual data points to show the range of values along with the direction and magnitude of change at the individual level rather than solely presenting the summary dynamite plunger which can often be misleading (e.g., Drummond & Vowler (2011) https://doi.org/10.1113/expphysiol.2011.057323). Consequently, we believe that the presentation of Figure 3 captures our data appropriately to enable readers to interpret the changes in the mean appetite ratings during the OGTT.

The statistical data embedded in Figure 3 represents the post-hoc comparison of the diet-by-time interactions. We decided to run the post-hoc tests by examining the magnitude of the pre-to-post change in appetite perceptions within each diet separately. Given the crossover design of the study and the 3 week washout between diets, we did not anticipate between-diet differences in the appetite responses at baseline (i.e., pre-diet) and we felt the magnitude of change over time within each diet (i.e., the pre-to-post change) would provide the most meaningful interpretation of the interaction effects. Indeed, the interaction effects for the appetite perceptions are partly driven by the pre-to-post change in the appetite perceptions occurring in opposite directions for the two diets. For example, there is a small increase in hunger in response to the control diet whereas hunger is supressed in response to the HE-HFD.  

When the post-hoc tests are constructed to compare the appetite responses between the two diets directly at each timepoint individually, there was no statistical between-diet differences in any of the appetite perceptions at the pre-diet time-point (all P ≥ 0.428). In contrast, at the post-diet timepoint on day 7, the values were lower in the HE-HFD vs control diet for hunger (mean difference (95% CI) -8 (-15, -2) mm, P = 0.015), desire to eat (-7 (-13, -1), P = 0.032), and PFC (-8 (-14, -1), P = 0.035), and higher in the HE-HFD vs control diet for fullness (10 (3, 17) mm, P = 0.004).    

Consequently, the interpretation of the interactions is the same irrespective of whether the post-hoc analysis is anchored by time or diet. Specifically, the interactions demonstrate that the HE-HFD reduced mean ratings of hunger, desire to eat and PFC, and increased mean ratings of fullness (as described on page 6, lines 206-208). This is consistent with the data depicted in Figure 3 and we are confident the interpretation of the interactions in the results and discussion is an accurate reflection of the data.

Reviewer comment 5: I think the authors are a bit enthusiastic about the results of the Leeds food Preference questionnaire presented in table 2. The conclusion is a little strong saying that the HE-HFD evoked a rapid reduction in the participants' desire and motivation to consume high-fat foods.

Authors’ response: We agree that the effects of the HE-HFD on hedonic outcomes should not be overstated because the magnitude of change was subtle. With regards to the text highlighted in this comment (by the reviewer), we feel that it is important to note the sentence that follows which reads: ‘These alterations in subjective, hormonal, and hedonic indicators of appetite were relatively modest and of small-to-moderate magnitude, despite the considerable energy surplus delivered in our overfeeding model.’ This sentence therefore qualifies the statement we are making. Nonetheless, we have subtlety changed the terminology used on lines 274-275 to ensure the findings are not overstated.

We have also reviewed the main body of text, and formal conclusion, and believe that we have adequately qualified statements about the magnitude of effect regarding LFPQ outcomes.

Reviewer 2 Report

This is a well-written paper that quantifies the effect between overfeeding and appetite-related hormones, and presents a sound and comprehensive discussion on the contribution of research findings and limitations. However, there are a few weakness listed as follows:

1) Figures of low-resolution lowered the quality of result presentation.

2) Too coarse measurements of energy intake from self-reports. The experiment design relies on self-reported measures, which have been widely known as unreliable for research settings due to frequently observed over-reported/under-reported issues.

3) Energy expenditure is not quantified. In the experiment design, the baseline (required) energy intake is estimated as BMR *1.7 (physical activity) *1.1 (thermic effect of food).

 3.a) Firstly, BMR equation being used is not provided

 3.b) Secondly, how was 1.7 determined? What activity level is expected with 1.7

 3.c) Lastly, non-exercise activity thermogenesis accounts for around 10% of total energy expenditure, and may be even higher in over-feeding subjects. This is an important expenditure components that help balance the energy in case of overfeeding, but was missing in current study for baseline energy requirement estimation.

4) It would be interesting to track with a follow-up measurement of weight gain on over-feeding to validation the impact from changes in appetite hormones.

Author Response

Reviewer 2

Reviewer comment 1: This is a well-written paper that quantifies the effect between overfeeding and appetite-related hormones, and presents a sound and comprehensive discussion on the contribution of research findings and limitations. However, there are a few weakness listed as follows:

Authors’ response: We thank the reviewer for their positive comments.

Reviewer comment 2: Figures of low-resolution lowered the quality of result presentation.

Authors’ response: We have now corrected this.

Reviewer comment 3: Too coarse measurements of energy intake from self-reports. The experiment design relies on self-reported measures, which have been widely known as unreliable for research settings due to frequently observed over-reported/under-reported issues.

Authors’ response: We agree with the reviewer about the limitations of self-reported energy intake data. In this study, three day weighed food records were completed at baseline (pre-intervention) and during the seven-day control phase in order to assess whether participants were following their habitual diet within this phase of the study. We accept that there are limitations in the accuracy of these data; and that a more robust method would have been to provide all food and drink for participants during the control phase (as we did during the high-fat diet phase). Unfortunately, this was not possible because of funding constraints, but ultimately, we are confident that the magnitude of overfeed would be considerable enough in the high-fat diet arm to test our hypotheses.

To highlight this limitation, we have now altered a sentence in our discussion (page 13, lines 389-392) which now reads:

‘The robust standardisation of dietary and experimental procedures is a key strength of this study; however, it is recognised that the experimental design could have been strengthen by the provision of all foods to participants in both trial arms (not just in the HE-HFD arm).’

Reviewer comment 4: Energy expenditure is not quantified. In the experiment design, the baseline (required) energy intake is estimated as BMR *1.7 (physical activity) *1.1 (thermic effect of food).

Firstly, BMR equation being used is not provided

Author’s response: The following equation has been added to lines 125-126: (resting energy expenditure = 10 x weight [kg] + 6.25 x height [cm] – 5 x age [years] + 5)

Secondly, how was 1.7 determined? What activity level is expected with 1.7

Author’s response: A PAL of 1.7 is considered a moderate activity level which was consistent with our participant group (healthy young adults who were recreational exercisers) (http://www.fao.org/3/y5686e/y5686e07.htm#bm07.3).

We have added detail to the sentence on line 127 to clarify this which now reads: ‘This value was multiplied by a physical activity correction factor that represents ‘moderate’ habitual activity levels (1.7) and uplifted by 10% to adjust for the thermic effect feeding.’

Lastly, non-exercise activity thermogenesis accounts for around 10% of total energy expenditure, and may be even higher in over-feeding subjects. This is an important expenditure components that help balance the energy in case of overfeeding, but was missing in current study for baseline energy requirement estimation.

Author’s response: this is a good point raised by the reviewer. Unfortunately, we did not include this factor within our calculation. In a practical sense, participants can tolerate our HE-HFD for seven days but not much longer. If the energy level had been increased further then we do not believe the diet would be tolerable. Ultimately, participants received a significant energetic stress (excess) that we believe was suitable to test our hypotheses.

Reviewer comment 5: It would be interesting to track with a follow-up measurement of weight gain on over-feeding to validation the impact from changes in appetite hormones.

Authors’ response: We agree, and one area that we would like to explore further is how long it takes appetite changes to reverse after overfeeding. To do this we would need additional assessments once participants had returned to their habitual diet.

With regards to weight gain, in this study, mean (SD) body mass pre and post the HE-HFD was 76.8 ± 3.7 and 78.0 ± 4.1 kg, respectively. These data were published along with our primary outcome data in a previous manuscript – Willis et al. (2020) https://doi.org/10.1093/jn/nxz333. Therefore, due to scientific integrity, we cannot use these data in the current manuscript.